# SELF-EVOLVING CURRICULUM FOR LLM REASONING

## ABSTRACT

Reinforcement learning (RL) has proven effective for fine-tuning large language models (LLMs), significantly enhancing their reasoning abilities in domains such as mathematics and code generation. A crucial factor influencing RL fine-tuning success is the training curriculum: the order in which training problems are presented. While random curricula serve as common baselines, they remain suboptimal; manually designed curricula often rely heavily on heuristics, and online filtering methods can be computationally prohibitive. To address these limitations, we propose *Self-Evolving Curriculum (SEC)*, an automatic curriculum learning method that learns a curriculum policy concurrently with the RL fine-tuning process. Our approach formulates curriculum selection as a non-stationary Multi-Armed Bandit problem, treating each problem category (e.g., difficulty level or problem type) as an individual arm. We leverage the absolute advantage from policy gradient methods as a proxy measure for immediate learning gain. At each training step, the curriculum policy selects categories to maximize this reward signal and is updated using the TD(0) method. Across three distinct reasoning domains: planning, inductive reasoning, and mathematics, our experiments demonstrate that SEC significantly improves models' reasoning capabilities, enabling better generalization to harder, out-of-distribution test problems. Additionally, our approach achieves better skill balance when fine-tuning simultaneously on multiple reasoning domains. These findings highlight SEC as a promising strategy for RL fine-tuning of LLMs. [1]

## 1 INTRODUCTION

Reinforcement learning (RL) has emerged as a central technique for fine-tuning large language models (LLMs) (Lightman et al., 2023; OpenAI; DeepSeek-AI, 2025), significantly improving their reasoning capabilities. Recent advances demonstrate notable success, particularly in domains where verifying generation correctness is straightforward (Lambert et al., 2024), such as mathematics and code generation. By optimizing LLMs with rewards solely defined by verifiable outcomes, RL fine-tuning encourages the emergence of complex reasoning behaviors, including self-correction and back-tracking strategies (Kumar et al., 2024; Yeo et al., 2025; Gandhi et al., 2025), that substantially enhance reasoning performance.

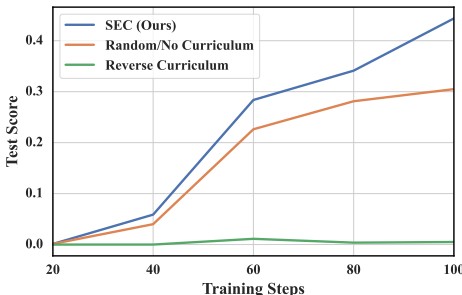

Figure 1: **Curriculum matters.** A deliberately poor (reverse) curriculum severely limits RL fine-tuning performance. Our proposed *Self-Evolving Curriculum (SEC)* significantly outperforms the standard random curriculum. See Sec. 3.1 for details.

A critical factor influencing the effectiveness of RL fine-tuning is the training curriculum (Bengio et al., 2009), i.e., the order in which training data is presented. Since online RL inherently depends on the policy model itself to produce high-quality training trajectories, aligning the curriculum with the model's current learning progress is critical. Ideally, such an alignment enables the model to continually encounter problems that yield maximal learning outcomes (Schaul et al., 2015; Loshchilov and Hutter, 2015; Katharopoulos and Fleuret, 2018). To illustrate this point concretely, we conduct a

---

[1]Our code will be open-sourced upon publication.

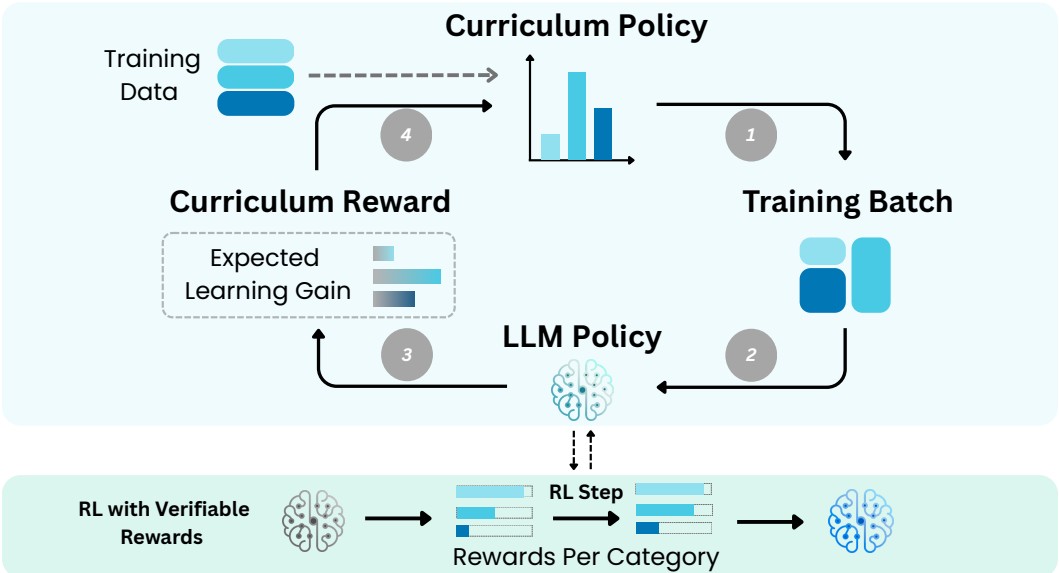

Figure 2: **Overview of *Self-Evolving Curriculum (SEC)*.** SEC dynamically adjusts the training curriculum according to the model's current capabilities. During preprocessing, training data is partitioned into distinct categories (indicated by colors), e.g., by difficulty level or problem type. At each RL fine-tuning step: (1) The curriculum policy samples a training batch based on categories' expected learning gains; (2) The LLM policy is updated using the sampled batch and the chosen RL algorithm; (3) Rewards for curriculum categories are computed using advantage values estimated by the RL algorithm; (4) The curriculum policy is updated accordingly, refining future data selection.

controlled experiment using the Countdown game,[2] deliberately employing a suboptimal (reverse) curriculum, in which problems are arranged from hard to easy. As shown in Figure 1, the resulting model performs poorly on the test set and exhibits minimal generalization to more challenging out-of-distribution (OOD) problems. In contrast, when trained with a baseline random curriculum, where problems of varying difficulty are drawn uniformly at random, the model demonstrates significantly improved generalization and overall task performance.

Although random curriculum serves as a reasonable baseline, it naturally raises the question: *Can we design more effective curriculum strategies*? Curriculum learning (Bengio et al., 2009) addresses precisely this challenge by seeking to optimize the sequencing of training tasks, thereby enhancing learning efficiency and efficacy. Recent approaches to curriculum learning for RL fine-tuning typically involve either manually crafted curricula designed upfront (Kimi Team et al., 2025; Song et al., 2025; Wen et al., 2025) or dynamic online filtering based on on-policy samples (Yu et al., 2025; Bae et al., 2025). However, manually designed curricula rely heavily on heuristics, demanding human intervention for new models or tasks; conversely, online filtering methods incur substantial computational overhead due to the continuous generation of additional on-policy samples.

In this paper, we propose *Self-Evolving Curriculum* (SEC) (Figure 2), an automatic curriculum learning (Portelas et al., 2020) approach for RL fine-tuning of LLMs. Our method adaptively learns a curriculum policy concurrently with the RL fine-tuning process, formulating curriculum selection as a non-stationary Multi-Armed Bandit (MAB) problem (Thompson, 1933; Sutton and Barto, 2018; Matiisen et al., 2020). Each curriculum category (e.g., difficulty level or problem type) is treated as an individual arm, and the curriculum policy aims to select the arm that maximizes the learning outcomes. Specifically, we operationalize the concept of learning outcomes using the gradient norm, noting that, in policy gradient methods, the gradient norm is weighted by the absolute value of the advantage function. Leveraging this observation, we define the absolute advantage as the reward for each arm. At each RL training step, the curriculum is sampled according to the current MAB policy, which is subsequently updated on-the-fly using the reward obtained from the current training step via the TD(0) method (Sutton, 1988).

---

[2]A puzzle game where players combine a given set of numbers using basic arithmetic operations to reach a target number.

Our experiments demonstrate that SEC significantly improves model reasoning capabilities across three distinct domains: planning, inductive reasoning, and mathematics, particularly improving generalization to challenging out-of-distribution problems. Compared to the standard random curriculum, SEC achieves substantial relative improvements, such as 13% on Countdown, 21% on Zebra puzzles, 22% on ARC-1D, and up to 33% on the AIME24 dataset. When fine-tuned simultaneously across multiple reasoning domains, SEC effectively balances performance across tasks, underscoring its strength as an automatic curriculum learning strategy for RL fine-tuning of LLMs.

## 2 METHOD

In the context of RL fine-tuning, at each training step $t$, the curriculum policy selects a subset $D_t \subseteq D$ from the training problem set $D$ to be provided to the LLM. In our work, we consider scenarios where the training problems can be categorized into $N$ distinct categories. This assumption simplifies the curriculum optimization problem into learning an expected return $Q_t(c)$ that maps category $c$ to a real-valued score (Sec. 2.1). The training batch is then constructed by first sampling categories according to the curriculum policy, followed by sampling problems uniformly within the categories.

The goal of the curriculum policy is to maximize the LLM's final task performance. However, directly evaluating such performance would require completing the entire RL fine-tuning process, while the curriculum policy is better to be updated along with the training steps. To resolve this, we introduce a locally measurable reward as a proxy objective for guiding the curriculum policy (Sec. 2.2).

### 2.1 CURRICULUM SELECTION AS MULTI-ARMED BANDIT

Training datasets used for reasoning tasks can often be naturally decomposed into distinct categories. For example, if the dataset spans various reasoning domains, such as mathematics, coding, and planning, these domains naturally form distinct categories. When the dataset is homogeneous in task type or domain, a curriculum can still be constructed by categorizing examples based on in-domain levels, such as difficulty. For instance, the MATH dataset (Hendrycks et al., 2021) categorizes problems into five distinct difficulty levels based on the guidelines provided by Art of Problem Solving (AoPS). Furthermore, in the absence of explicit difficulty annotations, problem difficulty can be estimated by either using the empirical accuracy of the training LLM or prompting an expert LLM in an additional preprocessing step, as demonstrated by Shi et al. (2025).

Motivated by these considerations, we assume that, particularly for reasoning-focused datasets, training problems can be partitioned into $N$ distinct categories $C = \{c_1, c_2, \ldots, c_N\}$. Conceptually, the curriculum policy optimization problem can then be viewed as a partially observable Markov decision process (POMDP): the state corresponds to the current LLM policy, actions correspond to curriculum selection, and rewards are defined by observable performance metrics, such as the on-policy performance associated with the selected curriculum.

This POMDP formulation naturally resembles a non-stationary MAB problem, a connection also highlighted by Matiisen et al. (2020), where each arm represents a problem category $c_i$, and the objective is to learn the expected return $Q_t(c)$ associated with selecting category $c$ at training step $t$. Importantly, the MAB in this context is non-stationary: the expected reward distribution for each arm shifts as the LLM policy is updated over the course of training. To address this well-studied non-stationary bandit problem (Thompson, 1933; Sutton and Barto, 2018), we leverage the classic TD(0) method (Sutton, 1988) to iteratively update $Q_t(c)$:

$$Q_{t+1}(c) = \alpha r_t(c) + (1 - \alpha)Q_t(c), \tag{1}$$

where $\alpha$ is the learning rate, $Q_0(c) = 0$ initializes the scores to zero, and $r_t(c)$ denotes the reward defined in the next section. Note that this is also known as the Exponential Moving Average. The curriculum policy can then be simply defined over $Q_t(c)$, as elaborated in Sec. 2.3.

### 2.2 MEASURING LEARNING OUTCOMES WITH ABSOLUTE ADVANTAGE

An ideal curriculum should maximize the LLM's final performance on the test data after an entire training episode. However, directly measuring this objective requires completing a full RL fine-tuning cycle. Although evaluating intermediate checkpoints can partially mitigate this issue, frequent

evaluations are computationally expensive. To overcome this challenge, we introduce a proxy objective that can be efficiently computed locally at each training step.

An intuitive choice for such a proxy objective is to prioritize training data that maximizes the model's immediate learning outcomes (Schaul et al., 2015; Loshchilov and Hutter, 2015; Katharopoulos and Fleuret, 2018), i.e., data that induces large parameter updates. Practically, this can be quantified by measuring the gradient norm of the loss function with respect to the selected training data. Specifically, consider a policy gradient algorithm that optimizes the LLM policy by minimizing the following loss function:

$$\mathcal{L}_{\text{PG}}(\theta) = - \mathbb{E}_{(s_t, a_t) \sim \pi_\theta} \big[ \log \pi_\theta(a_t \mid s_t) \, \widehat{A}_t \big] \tag{2}$$

where $\pi_\theta$ denotes the LLM policy and $\widehat{A}_t$ denotes the advantage value. Then, the per-step $(s_t, a_t)$ gradient norm is:

$$\|\nabla_\theta \mathcal{L}_{\text{PG}}(\theta, s_t, a_t)\|_2 = \left\| \mathbb{E}_{(s_t, a_t) \sim \pi_\theta} \left[ \nabla_\theta \log \pi_\theta(a_t \mid s_t) \, \widehat{A}_t \right] \right\|_2 \approx |\widehat{A}_t| \| \nabla_\theta \log \pi_\theta(a_t \mid s_t) \|_2$$

We observe that the gradient magnitude is weighted by the absolute value of the advantage $|\widehat{A}_t|$. We therefore approximate the learning gain of a curriculum $c$ by the batch-wise expectation of $|\widehat{A}_t|$:

$$r(c) = \mathbb{E}_{(s_t, a_t) \sim \pi_\theta(x_i), x_i \sim c} |\widehat{A}_t| \tag{3}$$

In other words, the reward for curriculum $c$ at each training step is computed as the average absolute advantage across all rollouts associated with the problems drawn from curriculum category $c$.

## 2.3 Self-evolving Curriculum for RL Fine-tuning

At each RL training step, a batch of problems is generated as follows. First, categories are sampled according to a Boltzmann distribution defined by the current values of $Q_t(c)$: $p(c) = \frac{e^{Q_t(c)/\tau}}{\sum_{i=1}^{N} e^{Q_t(c_i)/\tau}}$, where $\tau$ is the temperature parameter controlling the exploration-exploitation trade-off. Next, problems are uniformly sampled from the selected categories. This process is repeated until the desired batch size is reached. Sampling from the Boltzmann distribution naturally balances exploration and exploitation in curriculum selection.

The resulting batch is then used to update the LLM policy. After the policy update at each step, we compute the reward $r(c)$ for each sampled category $c$ and update the corresponding $Q_t(c)$ values using Eq. 1. The complete procedure of SEC is summarized in Algorithm 1.

## 3 Experiments

This section presents experiments evaluating SEC across three reasoning domains: **planning**, **inductive reasoning**, and **mathematics**. We additionally investigate SEC's effectiveness with different curriculum categories and alternative RL algorithms.

## 3.1 Experimental Setup

We conduct our experiments using the open-weight Qwen2.5 models (Yang et al., 2024): Qwen2.5-3B and Qwen2.5-7B. For RL fine-tuning on reasoning tasks, we employ the widely-used GRPO algorithm (Shao et al., 2024; DeepSeek-AI, 2025). We report average `pass@1` accuracy from the best checkpoint, calculated over 8 independent generations per problem. Additional training and evaluation details are provided in Appendix B. Prompts and data examples for all tasks are provided in Appendix C.

Our experiments cover three reasoning domains that require different abilities: (i) **Planning**, which requires look-ahead search and backtracking; (ii) **Inductive reasoning**, which involves learning general rules from observations and applying them to unseen scenarios; and (iii) **Mathematics**, which demands multi-step logical deduction and systematic problem solving.

---

**Algorithm 1** SEC: RL Fine-tuning with Self-evolving Curriculum

---

**Require:** Training set $D$ partitioned into categories $C = \{c_1, \ldots, c_N\}$; LLM policy $\pi_\theta$ with parameters $\theta$; Learning rate $\alpha$ (for $Q$ updates); Temperature $\tau$; Batch size $B$; Total training steps $T$; Reward function $\mathcal{R}$; RL algorithm $\mathcal{A}$
1: Initialize $Q_0(c) \leftarrow 0 \ \forall c \in C$
2: **for** $t \leftarrow 0$ **to** $T - 1$ **do**
3:     $B_t \leftarrow \emptyset$
4:     **while** $|B_t| < B$ **do**
5:         Sample category $c \sim \text{Softmax}\big(Q_t(c)/\tau\big)$
6:         Sample problem $x$ **uniformly** from category $c$
7:         $B_t \leftarrow B_t \cup \{x\}$
8:     **end while**
9:     Run $\pi_\theta$ on each $x \in B_t$ to generate rollouts $\mathcal{T}$ and compute rewards $\mathbf{r}$ with $\mathcal{R}$
10:    Estimate advantages $\widehat{A}$ and update $\pi_\theta$ with $\mathcal{A}(\pi_\theta, \mathcal{T}, \mathbf{r})$
11:    **for all** $c \in C$ **do**
12:       $B_c \leftarrow \{\, x \in B_t \mid x \text{ belongs to category } c \,\}$
13:       $r_t(c) \leftarrow \dfrac{1}{|B_c|} \sum_{j:x_j \in B_c} \dfrac{1}{T_j} \sum_t^{T_j} |\widehat{A}_{t,j}|$
14:       $Q_{t+1}(c) \leftarrow \alpha\, r_t(c) + (1 - \alpha)\, Q_t(c)$
15:    **end for**
16: **end for**
17: **return** Fine-tuned LLM $\pi_\theta$

---

**Planning.** For planning tasks, we consider two popular puzzle problems: (i) *Countdown*, where the goal is to use basic arithmetic operations to reach a target number from a given set of 3–6 integers. In this puzzle, we control the task difficulty by increasing the number of given integers. (ii) *Zebra Puzzles*, a classic logic puzzle involving 3–6 entities (e.g., houses) each with 3-6 properties (e.g., color). Given a set of textual clues (constraints), the goal is to correctly assign each property to each entity. Here, we control the task difficulty by increasing the number of entities and properties.

**Inductive reasoning.** We adopt the 1D variant of the Abstraction and Reasoning Corpus (ARC) (Chollet, 2019; Xu et al., 2023) for inductive reasoning. Each puzzle instance consists of strings of lengths 10, 20, 30, or 40 (with greater length corresponding to increased difficulty), which are defined over integers. Three input-output examples illustrating an underlying rule are provided, and the LLM is tested on an unseen case requiring generalization.

For the above three reasoning tasks (Countdown, Zebra, and ARC), we generate problems using the framework provided by Open-Thought (2025). Specifically, our training data consists of the three easiest difficulty levels, and the most difficult level is reserved as an out-of-distribution (OOD) evaluation set. For each difficulty level, we sample 10,000 problems for training and 200 held-out samples for evaluation. During RL fine-tuning, we assign rewards of 1 for correct problems, 0.1 for incorrect answers but with correct formatting, and 0 otherwise.

**Mathematics.** We train LLMs on the training split of the MATH dataset (Hendrycks et al., 2021), which comprises problems categorized into five difficulty levels, from 1 (easiest) to 5 (hardest), as specified in the dataset annotations. Unlike the previous three tasks, the training data for mathematics is imbalanced across these difficulty levels (Figure S1). For this task, we use a binary reward (1 for correct and 0 otherwise), without assigning a partial reward for a correct format. The models are subsequently evaluated on the MATH500, AMC22-23, and AIME24 datasets.

## 3.2 MAIN RESULTS

First, we evaluate the effectiveness of SEC using problem difficulty as the curriculum category, i.e., each difficulty level corresponds to an arm in the MAB framework. We compare SEC against two commonly used curriculum strategies: (1) *Random/No Curriculum*, where training samples are drawn uniformly across all difficulty levels following the original data distribution, corresponding to the conventional "no-curriculum" approach, and (2) *Difficulty-Ordered Curriculum*, where problems are

Table 1: **Evaluation across reasoning tasks and curriculum methods.** Accuracy is measured by averaging `pass@1` over 8 independent generations per problem. In-distribution (ID) results are averaged over test problems sampled from the same three difficulty levels used in training. The best-performing curriculum strategy for each dataset and model size is shown in **bold**, and the second-best is underlined. SEC consistently achieves strong performance across tasks, particularly improving generalization on challenging OOD test problems.

| Task | Split | Qwen2.5 3B | | | Qwen2.5 7B | | |
|---|---|---|---|---|---|---|---|
| | | **Random** | **Ordered** | **SEC (Ours)** | **Random** | **Ordered** | **SEC (Ours)** |
| *Countdown* | ID | 0.859 | 0.551 | **0.866** | 0.858 | 0.820 | **0.872** |
| | OOD | 0.479 | 0.321 | **0.542** | **0.566** | 0.442 | 0.555 |
| *Zebra* | ID | 0.517 | 0.534 | **0.547** | 0.573 | 0.572 | **0.587** |
| | OOD | 0.285 | 0.329 | **0.345** | 0.321 | 0.311 | **0.355** |
| *ARC-1D* | ID | **0.501** | 0.476 | 0.500 | 0.512 | **0.526** | 0.514 |
| | OOD | 0.313 | 0.363 | **0.381** | **0.436** | 0.428 | 0.418 |
| *Math* | MATH500 | 0.668 | **0.672** | **0.672** | **0.774** | 0.759 | 0.761 |
| | AMC22-23 | 0.345 | **0.352** | 0.351 | 0.486 | 0.477 | **0.511** |
| | AIME24 | 0.075 | 0.054 | **0.100** | 0.138 | 0.150 | **0.175** |

sequentially presented from easiest to hardest. Hyperparameters for SEC across all experimental settings are detailed in Appendix B.

Our results, summarized in Table 1, demonstrate clear advantages of SEC across tasks and models. For the smaller Qwen2.5-3B model, SEC consistently achieves substantial improvements on harder, out-of-distribution (OOD) test sets. Specifically, on Countdown, SEC significantly improves OOD accuracy by approximately 13% relative ($0.48 \rightarrow 0.54$) compared to the random baseline, and by approximately 69% relative ($0.32 \rightarrow 0.54$) compared to the difficulty-ordered baseline. Similarly, on Zebra, SEC attains a relative improvement of approximately 21% over random ($0.29 \rightarrow 0.35$). In mathematics, SEC markedly improves performance on the challenging AIME dataset by approximately 33% relative compared to the random baseline ($0.075 \rightarrow 0.10$).

For the larger Qwen2.5-7B model, SEC performance is competitive but more similar to the random curriculum on tasks like Countdown and ARC. This outcome aligns with expectations, as stronger base models may already possess sufficient reasoning capabilities to tackle harder problems, thus rendering explicit curriculum guidance less critical. Nevertheless, on more challenging tasks such as Zebra and mathematics, SEC continues to show clear improvements. Specifically, the OOD accuracy on Zebra improves by approximately 11% relative ($0.32 \rightarrow 0.36$) over the random baseline. On the challenging AIME problmes, SEC achieves a 27% relative gain ($0.14 \rightarrow 0.18$). The consistent improvements observed in these more challenging domains, together with the robust gains in the 3B model, highlight SEC's effectiveness in improving the model generalization.

For the larger Qwen2.5-7B model, SEC performance is competitive but closer to the random curriculum on tasks like Countdown and ARC. This outcome aligns with expectations: stronger base models often possess sufficient reasoning competence to tackle a broad range of problems, reducing the marginal benefit of an explicit curriculum. Nevertheless, on more challenging domains such as Zebra and mathematics, SEC continues to provide clear gains—e.g., a relative improvement of approximately 11% on Zebra OOD ($0.32 \rightarrow 0.36$) and 27% on AIME ($0.14 \rightarrow 0.18$).

To test whether curriculum benefits re-emerge as task difficulty increases, we evaluate Qwen2.5-7B on a harder Countdown variant (training on problem sizes 4–6 and evaluating OOD at size 7; previously 3–5 train and 6 OOD). As shown in Table 2, SEC again outperforms random on both ID and OOD, confirming that curriculum benefits persist for larger models when the task becomes more challenging.

Finally, the difficulty-ordered curriculum often yields suboptimal performance, likely due to its fixed difficulty schedule, which does not dynamically adapt to the model's current performance. As a result, models may spend excessive training time on easy problems, limiting exposure to harder ones from which models could potentially learn more. These results further underscore the necessity of adaptive, online curriculum strategies like SEC, which continuously align problem selection with the model's current learning state.

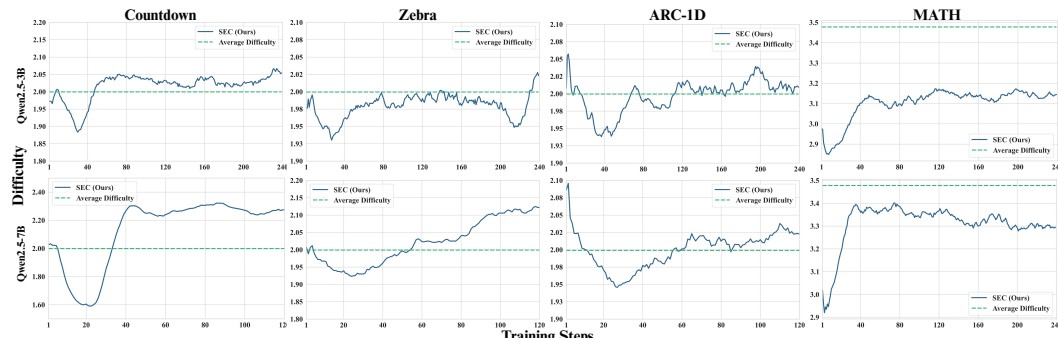

Figure 3: **Average sample difficulty over training steps.** SEC adaptively adjusts task difficulty during RL fine-tuning. Blue curves represent the sampled difficulty, smoothed using a moving average, while the green dashed line indicates the mean difficulty of the dataset. Across all benchmarks (columns) and model sizes (top: Qwen2.5-3B, bottom, Qwen2.5-7B), SEC initially selects easier problems and progressively introduces more challenging ones as training proceeds, effectively aligning difficulty with model improvement.

**Curriculum analysis.** Figure 3 illustrates how SEC adaptively adjusts training difficulty across tasks and models. For each task and model, the sampled difficulty (blue curves) initially starts below or around the dataset mean difficulty (green dashed line), indicating SEC's initial emphasis on easier problems to facilitate early-stage learning. As training progresses, SEC gradually increases the difficulty of selected problems, aligning the training complexity with the improving capabilities of the model. Notably, SEC selects harder problems for the stronger Qwen2.5-7B model compared to the smaller 3B model, further confirming SEC's ability to effectively adapt its curriculum to the model's learning capacity. This adaptive pattern across tasks and models highlights SEC's strength in dynamically adjusting problem difficulty to maximize learning outcomes.

Table 2: **Qwen2.5-7B on a harder Countdown variant.** SEC recovers clear gains over random when the task becomes more challenging.

| Method | ID | OOD |
|--------|-------|-------|
| Random | 0.654 | 0.373 |
| SEC | **0.686** | **0.439** |

## 3.3 SEC WITH MULTIPLE CURRICULUM CATEGORIES

In this section, we demonstrate that SEC seamlessly supports drawing from multiple and diverse curriculum categories at the same time. A common scenario in RL fine-tuning involves optimizing a model's performance across multiple reasoning domains. To evaluate SEC in such a multi-task setting, we combine the training datasets from Countdown, Zebra, and ARC to create a mixed training set comprising multiple types of reasoning problems, and conduct RL fine-tuning using the Qwen2.5-3B model. The goal here is to achieve a strong overall performance across all reasoning tasks.

Our MAB-based curriculum framework is agnostic to the semantic meaning of the curriculum categories, thus allowing categories to be defined arbitrarily. In this experiment, we define one arm for each unique combination of 3 problem types and 3 difficulty levels, resulting in a total of 9 distinct arms. We denote this variant as SEC-2D.

Figure 4 presents results evaluating SEC-2D when training simultaneously on multiple reasoning tasks. The table (left) demonstrates that SEC-2D consistently outperforms the random curriculum across all three reasoning tasks. The learning curve (right) provides a detailed view of OOD accuracy on Countdown as training progresses. Initially, both curricula show rapid improvement; however, the random curriculum exhibits a significant performance collapse midway through training, highlighting its inability to effectively balance multi-task learning. In contrast, SEC-2D maintains stable, robust performance, underscoring its strength in adaptively balancing multiple learning objectives.

Table 3: **SEC with alternative RL algorithms on Countdown.** SEC improves RL fine-tuning performance with different RL algorithms (PPO, RLOO), compared to a random curriculum.

| RL Method | Split | Random | SEC |
|-----------|-------|--------|-------|
| PPO | ID | 0.621 | **0.750** |
| | OOD | 0.159 | **0.224** |
| RLOO | ID | 0.821 | **0.859** |
| | OOD | 0.465 | **0.494** |

| Task | Split | Random | SEC-2D |
|------|-------|--------|--------|
| Countdown | ID | 0.837 | **0.839** |
|  | OOD | 0.418 | **0.428** |
| Zebra | ID | 0.513 | **0.539** |
|  | OOD | 0.254 | **0.312** |
| ARC | ID | 0.380 | **0.418** |
|  | OOD | 0.251 | **0.327** |

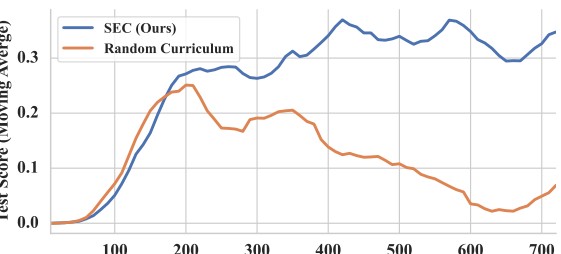

Figure 4: **Performance comparison when training on multiple tasks.** *Left:* Test accuracy of Qwen2.5-3B on ID and OOD splits. SEC-2D is implemented by defining an arm for each combination of problem type and difficulty level. SEC-2D consistently achieves higher accuracy, showing improved generalization compared to a random curriculum across tasks. *Right:* Countdown OOD accuracy vs. training steps, smoothed by a moving average. The random curriculum's performance collapses mid-training, highlighting its inability to effectively balance multiple tasks. In contrast, SEC-2D maintains stable performance throughout training.

### 3.4 SEC WITH ALTERNATIVE RL ALGORITHMS

While our main experiments employ the GRPO algorithm, we additionally evaluate SEC with other widely used RL methods, specifically Proximal Policy Optimization (PPO) (Schulman et al., 2017) and REINFORCE Leave-One-Out (RLOO) (Kool et al., 2019). Table 3 presents results on the Countdown task with Qwen2.5-3B, comparing SEC to the random curriculum under these two algorithms. Across both PPO and RLOO, SEC consistently improves performance on ID and OOD evaluation splits, demonstrating that it is effective beyond a single RL algorithm.

### 3.5 SEC WITH AUTOMATICALLY INFERRED CURRICULUM CATEGORIES

While our main experiments use curriculum categories predefined by metadata, SEC does not always require manual curation. To demonstrate this, we use the mathematics problem dataset released by Shi et al. (2025), which provides empirical success rates for each problem, estimated by sampling from Qwen2.5-MATH-7B. We discretize these success rates into $k$ equal-width bins to create curriculum categories (reporting results for $k=3$ and $k=5$), while keeping the RL setup, curriculum reward, and sampling policy unchanged.

Table 4 shows results on MATH500, AMC22–23, and AIME24. Both SEC variants outperform the random baseline on the three math benchmarks. These findings indicate that SEC can leverage automatically inferred curriculum categories derived from continuous difficulty signals, reducing reliance on manual labels.

Table 4: **Mathematics with automatically inferred curriculum categories.** SEC remains effective when curriculum categories are derived from empirical success rates rather than manual labels.

| Method | MATH500 | AMC22–23 | AIME24 |
|--------|---------|----------|--------|
| Random | 0.654 | 0.312 | 0.080 |
| SEC ($k=3$) | 0.664 | **0.389** | **0.088** |
| SEC ($k=5$) | **0.671** | 0.358 | 0.083 |

## 4 RELATED WORKS

**RL fine-tuning for language models.** Language models (LMs) can be naturally viewed as sequential decision-making policies, generating tokens conditioned on partial text states until reaching terminal outputs. Typically, reward signals are sparse and episodic, assigned only after the full generation, an approach termed Outcome Reward Models (ORM) (Cobbe et al., 2021). Some recent studies introduce Process Reward Models (PRM), assigning intermediate rewards during generation to facilitate local credit assignment (Lightman et al., 2023; Uesato et al., 2022). Leveraging this Markov Decision Process (MDP) framing, RL fine-tuning has demonstrated success across multiple domains, including aligning LMs with human preferences (RLHF) (Christiano et al., 2017; Ziegler et al., 2019; Bai et al., 2022; Ouyang et al., 2022), enhancing mathematical reasoning via exact-match rewards (Shao et al., 2024), and self-training with internal LM distributions (e.g., Self-taught Reasoner, STaR) (Zelikman et al., 2022). Recently, Reinforcement Learning with Verifiable

Rewards (RLVR) (DeepSeek-AI, 2025; Lambert et al., 2024) has emerged as a promising paradigm for improving the reasoning abilities of LMs.

RL methods tailored to these MDP formulations have also played a central role. Policy-gradient methods, including REINFORCE variants (e.g., RLOO) (Williams, 1992; Kool et al., 2019; Ahmadian et al., 2024) and Proximal Policy Optimization (PPO) approaches (Schulman et al., 2017; Shao et al., 2024), are widely adopted due to their relative stability. Alternatively, off-policy and value-based algorithms such as Directed Preference Optimization (DPO) (Rafailov et al., 2023; Meng et al., 2024) and Generative Flow Networks (GFlowNets) (Bengio et al., 2021; Hu et al., 2024; Ho et al., 2024) provide advantages in sample efficiency, diversity, and asynchronous training (Noukhovitch et al., 2024; Bartoldson et al., 2025), although they may not always match the task-specific reward maximization capabilities of on-policy methods, instead prioritizing improved diversity.

**Curriculum learning.** Curriculum learning was introduced by Bengio et al. (2009) and later refined as self-paced learning (Kumar et al., 2010), showing that organizing examples from easy to hard smooths non-convex optimization and improves generalization. In RL, curricula mitigate sparse rewards and exploration hurdles: reverse-curriculum generation grows start states outward from the goal (Florensa et al., 2017), Teacher-Student Curriculum Learning (TSCL) (Matiisen et al., 2020) also used a non-stationary MAB framework to maximize measured learning progress, defined as improvements in task performance, methods such as POET, ACCEL, and PAIRED co-evolve tasks with agents (Wang et al., 2019; Parker-Holder et al., 2022; Dennis et al., 2020), and Kim et al. (2025) proposed an adaptive teacher that dynamically adjusts curricula for multi-modal amortized sampling.

Only recently have similar curriculum learning ideas begun influencing RL fine-tuning of language models. $R^3$ applies reverse curricula specifically to chain-of-thought reasoning, progressively revealing longer reasoning sequences conditioned on gold demonstrations (Xi et al., 2024). Qi et al. (2024) proposed WEBRL, a self-evolving online curriculum RL framework designed to train LM-based web agents by prompting another LLM to autonomously generate new tasks based on previous failures.

Concurrently, several studies have explored automatic curriculum learning for RL fine-tuning. Bae et al. (2025) propose online filtering of training problems by repeatedly generating solutions to estimate their difficulty. AdaRFT (Shi et al., 2025) adaptively adjusts curriculum difficulty based on the model's recent reward signals but relies on explicit difficulty-level ordering. In contrast, SEC leverages a general MAB formulation to dynamically adjust the curriculum. DUMP (Wang et al., 2025b), in parallel to us, also leverages absolute advantage as the curriculum reward with an MAB framework, focusing on the Knights and Knaves logical reasoning puzzle with GRPO. In contrast, our study covers multiple reasoning domains, examines a multi-task RL setting, and validates performance across various RL algorithms.

## 5 CONCLUSION

In this paper, we introduced Self-Evolving Curriculum (SEC), an automatic curriculum learning framework tailored for RL fine-tuning of LLMs. SEC formulates adaptive curriculum selection as a non-stationary Multi-Armed Bandit problem, dynamically adjusting problem difficulty according to the model's evolving capability. Extensive experiments across diverse reasoning tasks, including planning, inductive reasoning, and mathematics, demonstrate that SEC consistently improves generalization and effectively balances learning across multiple reasoning domains simultaneously.

Our framework consists of three major components: curriculum rewards, sampling methods, and update rules. In this paper, SEC employs absolute advantage as the curriculum reward, a Boltzmann distribution for sampling, and a TD(0) update method. The generalization of these components can be explored for future work. For instance, one might incorporate uncertainty measures into the curriculum selection by leveraging approaches such as Upper Confidence Bound (UCB) (Auer, 2002) or Thompson sampling (Thompson, 1933).

**Limitations.** While SEC demonstrates consistent effectiveness across diverse reasoning tasks, it has some limitations. SEC introduces extra hyperparameters (e.g., temperature, learning rate) that require tuning. Future work may explore more flexible curriculum definitions, such as clustering problems based on embeddings or using lightweight models (e.g., linear regression) to directly estimate curriculum rewards.

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

## A  SEC ACROSS BASE MODEL FAMILIES

To examine whether SEC generalizes beyond the Qwen2.5 family, we fine-tune a Llama-3.2-1B variant released by Wang et al. (2025a), with speacial mid-training to improve RL fine-tuning performance. As shown in Table S1, one the Countdown task, SEC improves over the random curriculum on both splits, indicating that our method generalizes across model families and scales.

Table S1: **Llama-3.2-1B on Countdown.** SEC improves over random on both ID and OOD, suggesting generalization beyond a single model family.

| Method | Countdown ID | Countdown OOD |
|--------|--------------|---------------|
| Random | 0.758 | 0.192 |
| SEC | **0.776** | **0.265** |

## B  IMPLEMENTATION DETAILS

**Training.**    All models are fine-tuned with the GRPO algorithms (Shao et al., 2024) as implemented in the Volcano Engine Reinforcement Learning (verl) library (Sheng et al., 2024). We train separate 3B and 7B parameters variants of Qwen2.5 (Yang et al., 2024). The fine-tuning processes last in total 240 gradient steps for Qwen2.5-3B and 120 steps for Qwen2.5-7B with a batch size of 256 on each of the three puzzle tasks. Advantages are estimated by 8 rollouts. Both models are trained for 240 steps on the math task. We do not apply the Kullback-Leibler (KL) divergence loss by setting the corresponding loss weight to be 0 across our study. We limit the max prompt length to be 1,024 tokens and the max response length to be 4,096 tokens. The model parameters are optimized using Adam (Kingma and Ba, 2014) with a learning rate of 1e-6 and beta (0.9, 0.99) without warm-up steps. All of the training experiments are conducted on 4-8 NVIDIA H100 80GB GPUs. Hyperparameters for SEC used in each experiment is summarized in Table S2.

| Model | Countdown | Zebra | ARC | Math |
|-------|-----------|-------|-----|------|
| Qwen2.5 3B | $\alpha = 0.5$, $\tau = 1.0$ | $\alpha = 0.5$, $\tau = 1.0$ | $\alpha = 0.5$, $\tau = 1.0$ | $\alpha = 0.2$, $\tau = 1.0$ |
| Qwen2.5 7B | $\alpha = 0.5$, $\tau = 0.2$ | $\alpha = 0.5$, $\tau = 0.4$ | $\alpha = 0.5$, $\tau = 1.0$ | $\alpha = 0.5$, $\tau = 0.4$ |

Table S2: Hyperparameter settings (learning rate $\alpha$ and temperature $\tau$) used in each experiment.

For the multi-task experiment in Sec. 3.3, we fine-tune the Qwen2.5-3B model for $3 \times 240 = 720$ steps on the mixed dataset. We use $\alpha = 0.5$ and $\tau = 0.2$.

In Sec. 3.4, we train Qwen2.5-3B for 120 steps in all the experiments. For RLOO, we similarly use 8 rollouts for advantage estimation and $\alpha = 0.5$, $\tau = 0.25$ for SEC. For PPO, we use $\alpha = 0.5$, $\tau = 1$ for SEC, and $\lambda = 1$, $\gamma = 1$ for the GAE parameters. Consistent with our main experiments, we do not apply the KL divergence loss.

**Models.** Below we list the models used in our experiments:

- **Qwen2.5-3B:** https://huggingface.co/Qwen/Qwen2.5-3B

- **Qwen2.5-7B:** https://huggingface.co/Qwen/Qwen2.5-7B

**Math Datasets.** Below we list the data sources used in our experiments:

- **MATH500:** https://huggingface.co/datasets/HuggingFaceH4/MATH-500

- **AMC22-23:** https://huggingface.co/datasets/AI-MO/aimo-validation-amc

- **AIME:** https://huggingface.co/datasets/Maxwell-Jia/AIME_2024

## C   DATA EXAMPLES

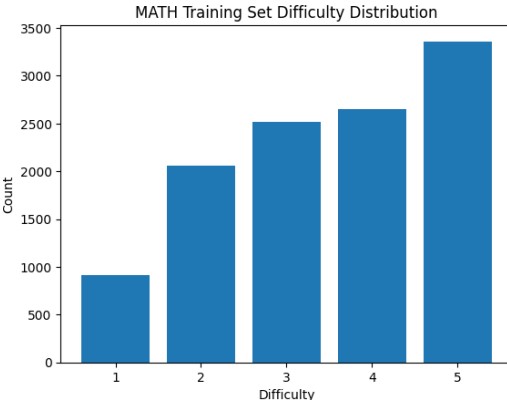

Figure S1: Distribution of difficulty levels in the MATH training set.

Below we list the prompts and data examples for each task in our study. The prompt template is adopted from Pan et al. (2025).

---

**Prompt for Countdown**:
```
A conversation between User and Assistant.  The user asks a question,
and the Assistant solves it.  The assistant first thinks about the
reasoning process in the mind and then provides the user with the
answer.
User:  Using the numbers [5, 17, 91], create an equation that equals
113.  You can only use basic arithmetic operations (+, -, *, /) and
each number should be used exactly once.  Return the final answer in
\boxed{}, for example \boxed{(1 + 2) / 3}.  Assistant:  Let me solve
this step by step.
```

---

**Prompt for Zebra Puzzle:**
```
A conversation between User and Assistant.  The user asks a question,
and the Assistant solves it.  The assistant first thinks about the
reasoning process in the mind and then provides the user with the
answer.
User:  This is a logic puzzle.  There are 3 houses (numbered 1 on the
left, 3 on the right), from the perspective of someone standing across
the street from them.  Each has a different person in them.  They have
different characteristics:
 - Each person has a unique name:  arnold, bob, alice
 - Everyone has a different favorite cigar:  dunhill, prince, pall mall
 - The people keep different animals:  cat, dog, bird

1.  The bird keeper is directly left of the Dunhill smoker.
2.  Alice is the dog owner.
3.  Arnold is in the second house.
4.  Alice is the Prince smoker.
5.  Arnold is the cat lover.

What is Name of the person who lives in House 1?  Provide only the
name of the person as your final answer and put in in \boxed{}, for
example: \boxed{Alice}.  Assistant:  Let me solve this step by step.
```

**Prompt for ARC-1D:**

A conversation between User and Assistant. The user asks a question, and the Assistant solves it. The assistant first thinks about the reasoning process in the mind and then provides the user with the answer.
User: Find the common rule that maps an input grid to an output grid, given the examples below.

Example 1:
Input: 1 2 1 2 1 0 0 1 2 0
Output: 0 0 0 1 1 1 1 2 2 2

Example 2:
Input: 1 2 0 0 0 0 2 0 1 2
Output: 0 0 0 0 0 1 1 2 2 2

Example 3:
Input: 0 0 2 0 0 0 0 1 1 0
Output: 0 0 0 0 0 0 0 1 1 2

Below is a test input grid. Predict the corresponding output grid by applying the rule you found. Describe how you derived the rule and your overall reasoning process in detail before you submit your answer. Your final answer must be placed in \boxed{} and should be just the test output grid itself.

Input: 0 0 2 0 0 1 1 0 0 1 Assistant: Let me solve this step by step.

**Prompt for math:**

A conversation between User and Assistant. The user asks a question, and the Assistant solves it. The assistant first thinks about the reasoning process in the mind and then provides the user with the answer.
User: Find the remainder when

$$33818^2 + 33819^2 + 33820^2 + 33821^2 + 33822^2$$

is divided by 17. Put your final answer within \boxed{}. Assistant: Let me solve this step by step.

