# OpenReview forum: "Self-Evolving Curriculum for LLM Reasoning"
_ICLR.cc/2026/Conference — ICLR 2026 Conference Withdrawn Submission_

### Official Review · Reviewer_6S7k · 2025-10-28

**Soundness:** 3
**Presentation:** 3
**Contribution:** 3
**Rating:** 4
**Confidence:** 4

**Summary:**

This paper introduces the Self-Evolving Curriculum (SEC), an automatic curriculum learning framework for fine-tuning Large Language Models (LLMs) with Reinforcement Learning (RL). The authors formulate curriculum selection as a non-stationary Multi-Armed Bandit (MAB) problem, where each "arm" corresponds to a category of problems. A key contribution is the use of the absolute advantage value from the policy gradient update as a computationally efficient proxy for immediate learning gain, which serves as the reward signal for the MAB. The arm selection policy is updated using the TD(0) method. Through extensive experiments on planning, inductive reasoning, and mathematics tasks with Qwen2.5 models, the paper demonstrates that SEC outperforms standard random and difficulty-ordered curricula.

**Strengths:**

1.  The motivation to frame the curriculum as a non-stationary MAB is intuitive. Using the absolute advantage as a reward signal is a natural choice.
2.  The experimental setup is thorough. The authors evaluate SEC across three distinct reasoning domains, using two different model sizes. Further ablations showing the method's effectiveness with different RL algorithms, and with automatically inferred curriculum categories strengthen the paper's claims.
3.  The results show the benefits of SEC, especially for improving generalization on out-of-distribution (OOD) test sets.

**Weaknesses:**

1.  The paper notes that the performance gap between the SEC and a random curriculum narrows for the larger Qwen2.5-7B model on several tasks. This raises an important question about the scalability of the approach's benefits. As foundation models become increasingly capable, the inherent need for a carefully curated curriculum might decrease, potentially limiting the long-term impact of this method.

2.  The paper employs a TD(0) update to handle the non-stationary nature of the MAB problem. While empirically successful here, this is a relatively simple approach that doesn't guarantee optimal adaptation. The non-stationary bandit literature offers more sophisticated algorithms (e.g., Sliding-Window Thompson Sampling[a]) that might provide more robust adaptation and stronger theoretical grounding.

a. Trovo, Francesco, Stefano Paladino, Marcello Restelli, and Nicola Gatti. "Sliding-window Thompson sampling for non-stationary settings." Journal of Artificial Intelligence Research 68 (2020): 311-364.

**Questions:**

1.  Following up on the observation that gains are smaller for the 7B model: How do you foresee SEC performing on even larger, state-of-the-art models (e.g., Qwen3-235B)?

2.  Could you comment on why TD(0) was chosen over other non-stationary bandit algorithms, such as sliding-window Thompson Sampling[a], which can offer stronger theoretical guarantees for adapting to distribution drift? Did you experiment with any alternatives?

a. Trovo, Francesco, Stefano Paladino, Marcello Restelli, and Nicola Gatti. "Sliding-window Thompson sampling for non-stationary settings." Journal of Artificial Intelligence Research 68 (2020): 311-364.

---

### Official Review · Reviewer_UpBW · 2025-10-28

**Soundness:** 2
**Presentation:** 3
**Contribution:** 2
**Rating:** 2
**Confidence:** 5

**Summary:**

The paper investigates reinforcement fine-tuning for reasoning models, aiming to improve training efficiency through better curriculum learning strategies. The authors highlight the limitations of existing approaches: random curricula (suboptimal), manually designed curricula (heuristic), and online filtering methods (computationally expensive). To address these issues, they propose an automatic curriculum framework (called SEC) that formulates curriculum selection as a non-stationary multi-armed bandit (MAB) problem. The SEC policy leverages absolute advantage as a reward function. Experiments on reasoning domains demonstrate that SEC improves reasoning capability and generalization to out-of-distribution (OOD) tasks.

**Strengths:**

- The paper is well-written and easy to follow.
- It tackles an important problem in fine-tuning reasoning models.
- The proposed solution is practical and has potential real-world impact.

**Weaknesses:**

The experimental comparison is limited to simple baselines (random and easy-to-hard curricula). However, several zone of proximal development (ZPD) and self-paced learning based curriculum RL methods (Florensa et al., 2018; Klink et al., 2020; Eimer et al., 2021; Tzannetos et al., 2023) exist with comparable computational cost to SEC. These approaches are also not discussed in the related work section.

In particular, ProCuRL (Tzannetos et al., 2023) provides a simple baseline. It applies to both binary and non-binary rewards and can be integrated into the paper's Algorithm 1 framework. Specifically, instead of Eq. (3), ProCuRL uses $r_t(c) = \mathbb{E}_{\tau \sim \pi_t(x), x \sim c}[R(\tau)]$ (where $R$ is the true reward function), with the curriculum policy defined as $p(c) \propto \exp (Q_t(c) \cdot (1 - Q_t(c)) / \tau)$. When the reward is non-binary, $Q_t(c)$ requires normalization. The computational cost of ProCuRL is comparable to SEC and could be considered as an additional baseline.

References:

Florensa et al., 2018: Automatic Goal Generation for Reinforcement Learning Agents.

Klink et al., 2020: Self-Paced Deep Reinforcement Learning.

Eimer et al., 2021: Self-Paced Context Evaluation for Contextual Reinforcement Learning.

Tzannetos et al., 2023: Proximal Curriculum for Reinforcement Learning Agents.

**Questions:**

It would be useful to report end-to-end wall-clock time comparisons across curriculum methods to assess computational overhead.

What is the novelty of SEC compared to existing MAB-based curriculum approaches such as Matiisen et al. (2020)?

In experiments, why is evaluation done using the best checkpoint rather than the last checkpoint? Does the curriculum cause any training instability?

---

### Official Review · Reviewer_XxBp · 2025-10-31

**Soundness:** 3
**Presentation:** 3
**Contribution:** 2
**Rating:** 2
**Confidence:** 4

**Summary:**

This paper introduces Self-Evolving Curriculum (SEC), a method that dynamically adjusts the training data sequence during Reinforcement Learning (RL) fine-tuning of Large Language Models (LLMs). The core idea is to formulate curriculum selection as a non-stationary Multi-Armed Bandit (MAB) problem, where each "arm" represents a category of problems (e.g., a difficulty level). The curriculum policy is updated concurrently with the RL policy using the average absolute advantage of a training batch as a reward signal, aiming to select data that maximizes immediate learning gain. Experiments across planning (Countdown, Zebra Puzzle), inductive reasoning (ARC-1D), and mathematics (MATH) domains demonstrate that SEC consistently improves generalization, particularly on out-of-distribution (OOD) tasks, compared to random or fixed-order curricula. The method also shows effectiveness in multi-task settings and with different RL algorithms.

**Strengths:**

1. Novelty: The formulation of adaptive curriculum learning as a non-stationary MAB problem is novel in the context of LLM RL-finetuning.

2. Strong Empirical Results: The paper provides extensive experiments across three distinct reasoning domains (planning, inductive reasoning, mathematics) and two model scales (3B and 7B parameters).

3. Clarity and Reproducibility: The paper is well-written, and Algorithm 1 provides a clear outline of the method. The authors have also included details on model variants, dataset sources, and hyperparameters, which aids reproducibility.

**Weaknesses:**

1. While the core method is well-evaluated, a more detailed ablation study would strengthen the paper. For instance, how crucial is the specific choice of the absolute advantage? How do the performance gains compare to the computational cost of maintaining and updating the MAB policy? Furthermore, the hyperparameters for the MAB (learning rate, temperature) are provided but not discussed in terms of their sensitivity or impact on final performance. A brief analysis would be valuable.

2. The connection between the absolute advantage and the actual learning progress is intuitively explained but lacks a rigorous theoretical justification. A more formal discussion or bound on how this proxy relates to the ultimate goal of maximizing final task performance would enhance the paper's foundation.

3. The method relies on a predefined or automatically inferred categorization of problem difficulty. The paper could more deeply discuss the limitations of this approach—for instance, how SEC might perform when "difficulty" is not easily quantifiable or when the model's perception of difficulty changes in non-monotonic ways during training.

**Questions:**

See Weaknesses

---

### Official Review · Reviewer_qVdB · 2025-11-03

**Soundness:** 3
**Presentation:** 3
**Contribution:** 2
**Rating:** 4
**Confidence:** 4

**Summary:**

This paper proposes an automatic curriculum learning method for large language models reinforcement learning. The curriculum signal is the batch average absolute advantage from the same on policy rollouts used for training, so no extra probing is required. Experiments on logic puzzles and mathematics with Qwen models at three and seven billion parameters show ID (in distribution) and OOD (out of distribution) gains over random or fixed curriculum.

**Strengths:**

* The studied problem is important.

* Different tasks (Inductive reasoning, Planning, and Math) are involved in the experiments.

* Both ID (in distribution) and OOD (out of distribution) settings are considered in the experiments.

* This paper is well-written.

**Weaknesses:**

* This paper does not compare against many existing curriculum-learning methods for reinforcement learning, despite a growing body of existing works (see references below). The related work section also listed many existing RL curriculum learning methods, but they are not compared in the experiments. The baselines used in the experiments are RFT method without curriculum or that with naive curriculum, which makes it hard to compare the proposed method and other advanced curriculum learning methods. A better evaluation should includes more existing RL curriculum learning methods.

* The high level idea that learning from easy to hard using a curriculum signal and formulating curriculum learning as a Multi-Armed Bandit problem is not new, as it's already commonly used in both traditional RL and LLM RL. What is the fundamental differences and advantages of the proposed method comparing to these existing works?

* The proposed method introduces multiple hyper-parameters. Based on Table S2, this paper has per-dataset hyperparameter choices, but does not include systematic sensitivity analyses or a clear selection protocol. Without sweeps or robustness tables, it is unclear whether the reported gains reflect the curriculum mechanism or tuning effects tailored to each dataset. Several key hyperparameters—especially decoding temperature—differ across datasets. Why are these settings not unified, and does this indicate the method is sensitive to hyperparameters? A stronger evaluation could justify how each hyperparameter was chosen (search vs. heuristic vs. validation tuning), report results under a single fixed configuration shared across datasets, and include sensitivity analysis.


Zhang et al., Learning Like Humans: Advancing LLM Reasoning Capabilities via Adaptive Difficulty Curriculum Learning and Expert-Guided Self-Reformulation. EMNLP 2025.

Tzannetos et al., Proximal Curriculum for Reinforcement Learning Agents. TMLR 2023.

Narvekar et al., Curriculum Learning for Reinforcement Learning Domains: A Framework and Survey. JMLR 2021.

Parashar et al., Curriculum Reinforcement Learning from Easy to Hard Tasks Improves LLM Reasoning

**Questions:**

see above

---

### Note · Authors · 2025-12-08

I have read and agree with the venue's withdrawal policy on behalf of myself and my co-authors.